# Central Retinal Artery Occlusion Is Related to Vascular Endothelial Injury and Left Ventricular Diastolic Dysfunction

**DOI:** 10.3390/jcm11082263

**Published:** 2022-04-18

**Authors:** Jerzy Dropiński, Radosław Dziedzic, Agnieszka Kubicka-Trząska, Bożena Romanowska-Dixon, Teresa Iwaniec, Lech Zaręba, Jan G. Bazan, Agnieszka Padjas, Stanisława Bazan-Socha

**Affiliations:** 1Jagiellonian University Medical College, Department of Internal Medicine, Faculty of Medicine, 31-066 Cracow, Poland; agnieszka.padjas@uj.edu.pl (A.P.); stanislawa.bazan-socha@uj.edu.pl (S.B.-S.); 2Jagiellonian University Medical College, Students’ Scientific Group of Immune Diseases and Hypercoagulation, 31-066 Cracow, Poland; radoslawjozefdziedzic@gmail.com; 3Jagiellonian University Medical College, Department of Ophthalmology and Clinic of Ophthalmology and Ocular Oncology of University Hospital, Faculty of Medicine, 31-501 Cracow, Poland; agnieszka.kubicka-trzaska@uj.edu.pl (A.K.-T.); bozena.romanowska-dixon@uj.edu.pl (B.R.-D.); 4Jagiellonian University Medical College, Department of Hematology, 31-501 Cracow, Poland; teresa.iwaniec@uj.edu.pl; 5University of Rzeszow, College of Natural Sciences, Interdisciplinary Center for Computational Modelling, 35-310 Rzeszow, Poland; lzareba@ur.edu.pl (L.Z.); jbazan@ur.ed.pl (J.G.B.)

**Keywords:** central retinal artery occlusion, ultrasonography, vascular endothelial dysfunction, flow-mediated dilatation, intima-media thickness

## Abstract

Central retinal artery occlusion (CRAO) is an emergency state characterized by sudden, painless vision impairment. Patients with CRAO have an increased risk of cardiovascular events, including stroke, likely related to vascular endothelial damage. Therefore, we investigated flow-mediated dilatation (FMD) of the brachial artery as a marker of endothelial dysfunction, intima-media complex thickness (IMT) of the common carotid artery, pointing to the arterial wall atherosclerotic alteration, and transthoracic echocardiographic parameters in 126 consecutive CRAO patients (66 men [52.4%], median age 55 years) and 107 control participants (56 men [52.3%], matched by age, sex, and body mass index). Most CRAO patients (*n* = 104, 82.5%) had at least one internal medicine comorbidity, mainly hypercholesterolemia and hypertension, which coexisted in one-fourth of them. Furthermore, they had a 38.2% lower relative increase of FMD (FMD%) and a 23.1% thicker IMT compared to the controls (*p* < 0.001, both, also after adjustment for potential confounders). On echocardiography, the CRAO group was characterized by increased dimensions of the left atrium and thicker left ventricular walls, together with impaired left ventricular diastolic function. CRAO is related to vascular endothelial damage, atherosclerosis, and left ventricular diastolic cardiac dysfunction. Thus, non-invasive ultrasound assessments, such as FMD%, IMT, and echocardiography, may be helpful in screening patients with increased CRAO risk, particularly those with other comorbidities.

## 1. Introduction

A central retinal artery occlusion (CRAO) is an ophthalmic emergency typified by a sudden, non-traumatic, and painless vision loss. The visual prognosis, although usually poor, varies depending on the etiology and duration of retinal ischemia [1]. Atherosclerosis-related thrombosis at the level of the lamina cribrosa is the most common cause of CRAO, accounting for about 80% of cases. Vasospasm and embolism are other but less frequent causes of CRAO. The CRAO incidence is estimated to be 1.9/100,000 person-years in the USA population, slightly more prevalent in males and rising with age, being about five times higher in those above 80 years of age [2,3].

Interestingly, CRAO is widely considered to be an ocular form of acute ischemic stroke, as follows: both have similar etiology, and in both, the pathogenesis is strongly associated with cardiovascular risk factors. Therefore, patients with CRAO should be assessed immediately in a stroke center to minimize the risk of further ophthalmic complications, such as blindness and irreversible retinal damage [4,5]. Additionally, as a meta-analysis by Fallico et al. [6] revealed, in one-third of patients within seven days after the episode of CRAO, the presence of signs of acute cerebral ischemia on magnetic resonance imaging is detected. Thus, CRAO should be considered not only as an ophthalmic emergency—threatening the vision—but also as a vascular event that may precede the development of life-threatening systemic vascular complications. Furthermore, the study performed by Shaikh et al. [7] showed that traditional cardiovascular risk factors, including advanced age, arterial hypertension, hyperlipidemia, and tobacco use, as well as non-stroke cerebrovascular disease and cardiac valvular illness, are associated with an increased stroke rate in patients with past CRAO episodes. Moreover, CRAO patients had a high occurrence of other cardiovascular complications, including subclinical coronary artery disease [8], atrial fibrillation [9], and valvular heart disease [10]. Thus, even though symptoms affect the eye, CRAO should be considered as a possible harbinger of other serious, potentially life-threatening cardiovascular diseases (CVD), including stroke or myocardial infarction [11,12].

Although the pathogenesis of CRAO is still not fully elucidated, several explanations have been suggested, related mainly to thromboembolic blockage of the central retinal artery or, less frequently, to vasospasm [13]. However, independently of the pathomechanism, the visual prognosis of ongoing CRAO is poor due to retinal infarction. An experimental transient CRAO evoked in rhesus monkeys demonstrated irreversible retinal changes after about 4 h of ischemia [14]. Furthermore, a study in pigs revealed that unalterable eye damage is preceded by potentially treatable local edema [15]. Therefore, early diagnosis and proper therapy are crucial for the visual outcome. Unfortunately, reliable biomarkers or screening tests to identify patients threatened with the first CRAO episode are unavailable. Some studies indicated a low-grade systemic inflammation in that disease; nevertheless, it was unspecific [12,16]. Furthermore, it has been shown that some genetic variants of interleukin(IL)-6 and IL-10 might be related to its pathogenesis, indicating once again the proinflammatory background of CRAO incidence [17,18].

Since typical risk factors of cardiovascular diseases, including classical, genetic, and metabolic ones, are involved in the pathogenesis of CRAO [19], they may also lead to endothelial dysfunction and early atherosclerosis [20,21]. Endothelium, by several regulatory substances, is maintaining the proper hemostatic balance, controlling the process of coagulation, and managing the vascular tone and blood vessels formation and growth. Thus, endothelium injury may lead to a prothrombotic and proinflammatory state and decreased vasodilatation [22]. The vascular endothelial function may be assessed best by invasive methods [23]. However, a non-invasive ultrasound examination based on brachial artery dilatation after hyperemia is also recommended for that purpose. In that test, shear stress leads to nitric oxide release and proper vasodilatation [24].

Considering the possible association between CRAO and vascular endothelial damage, we decided to measure the brachial artery flow-mediated dilatation (FMD) in those patients. We also analyzed intima-media complex thickness (IMT) in the common carotid artery, which can be a helpful tool in assessing the advancement of atherosclerosis, especially at the beginning of the process, as a CVD risk factor [25]. Furthermore, we performed standard transthoracic echocardiography, which is routinely advised to assess the cardiovascular system [26]. To date, several studies have demonstrated unfavorable decreased FMD and/or thicker IMT in some ophthalmic diseases, including glaucoma [27], diabetic retinopathy [28], and age-related macular degeneration [29]. However, no comprehensive data is available for the CRAO. It is worth emphasizing that this acute ophthalmic condition is still a significant diagnostic and therapeutic challenge for physicians due to various sudden clinical symptoms, complex etiology, and difficult therapeutic and prophylactic management. Thus, selecting a test or a biomarker that identifies individuals at increased CRAO risk would be beneficial. To the best of our knowledge, it is the first study on vascular endothelial dysfunction analyzed in CRAO patients.

## 2. Materials and Methods

### 2.1. Study Population

The study had a case-control, single-center format and the Bioethics Committee of the Jagiellonian University Medical College approval was obtained (permit No: KBET/79/B/2013). All procedures were carried out under the ethical guidelines of the Declaration of Helsinki. Patients were instructed about the purpose, methodology, and safety protocol of this study. Informed consent was obtained from all participants in this study.

We investigated 126 adult patients suffering from CRAO, 66 [52.4%] men, median age as follows: 55, range 18–80 years. We enrolled patients in the period from 2013 to 2019 at the Outpatient Clinic of the Department of Allergy and Clinical Immunology, University Hospital, Krakow, Poland, who were referred from the Department of Ophthalmology, Clinic of Ophthalmology and Ocular Oncology of the Jagiellonian University Medical College in Krakow, Poland, after an episode of CRAO. The diagnosis of CRAO was determined by an ophthalmologist based on the following typical symptoms: a history of sudden, painless visual loss in the involved eye, profound relative afferent pupillary defect, and the presence of characteristic findings on fundoscopy including ischemic retinal edema, a “cherry-red spot” sign in the macula, retinal arteriolar attenuation, segmentation of blood flow in retinal arterioles, and normal appearance of the optic disc [30]. The severity of the CRAO was established on the results of BCVA (best-corrected visual acuity) and fundus appearance (degree of arteriole attenuation and ischemic retina edema). A sudden visual loss was profound in all patients, ranging from counting fingers to light perception. No macular sparing due to the presence of a cilioretinal artery was observed in the analyzed group of patients. Additionally, patients in our study underwent diagnostic procedures for one to three months after the onset of the CRAO episode. Most of them were treated initially with low molecular weight heparins followed by low doses of aspirin. Only a minority of them were treated with heparin and aspirin together. Moreover, only seven (5.6%) of our patients received fibrinolytic therapy.

The control group comprised 107 healthy sex, age, and body mass index (BMI)-matched individuals, 56 men [52.3%], median age as follows: 53, range 34–78 years. Controls were recruited voluntarily from the hospital personnel and their relatives and had no history of thromboembolic diseases.

In both groups, exclusion criteria included the following: any acute illness during the last few months, history of myocardial infarction or stroke, ongoing cancer treatment, congestive heart failure, liver dysfunction, or renal insufficiency. Liver dysfunction was defined as an elevation of serum alanine aminotransferase more than twice above the upper limit of the reference range. Renal failure was described as an estimated glomerular filtration rate (eGFR) below 60 mL/min/1.73 m^2^. Patients with arterial hypertension, hypercholesterolemia, or diabetes mellitus were included in the study. Arterial hypertension was determined by a history of higher blood pressure (BP) than 140/90 mmHg or current antihypertensive treatment. Hypercholesterolemia was stated as serum total cholesterol more than 5.2 mmol/L or current antihypercholesterolemic therapy. Diabetes mellitus was defined as fasting serum glucose above 7.0 mmol/L or present use of insulin or hypoglycemic agents. Smoking habit was defined as the use of at least one cigarette daily. Family history of CVD was considered positive in patients with confirmed CVD in a first-degree relative.

### 2.2. Laboratory Analysis

Fasting blood samples were drawn in the morning from the ulnar vein with minimal tourniquet use after sufficient rest. Complete blood cell count, glucose, creatinine, and lipid profile were measured with routine laboratory techniques. The measurement of C-reactive protein (CRP) was performed with the Johnson & Johnson VITROS 250. The homocysteine level was evaluated with a chemiluminescent microparticle immunoassay. The blood samples were kept in tubes with 0.109 mol/l sodium citrate (*v*/*v*, 9:1) centrifuged at 2000× *g* for 10 min at room temperature within two hours of collection. The values of CRP < 5 mg/L and homocysteine < 16 μmol/L were considered normal.

### 2.3. Ultrasound Examinations

Ultrasound studies were performed on all enrolled individuals in a darkened, quiet room with subjects resting in a supine position after an adequate rest using the Siemens Acuson Sequoia 512 with a 10-MHz linear array ultrasonic transducer (Mountain View, CA, USA). Two ultrasound experts conducted these examinations independently of each other. The final result for each parameter was the mean of these measurements.

### 2.4. Flow-Mediated Dilatation of the Brachial Artery (FMD)

We measured FMD of the brachial artery using Celemajer’s method, as described previously [31]. Briefly, the baseline sagittal diameter (D1) of the distal part of the brachial artery was evaluated in the M-presentation with a 10 MHz linear array ultrasonic transducer placed approximately 3 cm proximal to the arterial bifurcation. After that, a sphygmomanometer cuff was placed on the forearm below the elbow, inflated to a pressure of 200 mmHg for 5 min, and then released. The brachial artery diameter was measured and re-recorded (D2) at the same point one minute after releasing the cuff. To clarify, FMD was described as a change in the increase in the brachial artery diameter after deflation of the cuff and was calculated as follows: FMD% = [(D2 − D1)/D1] × 100%.

### 2.5. Intima-Media Thickness (IMT) of the Common Carotid Artery

Measurements of intima-media thickness (IMT) of the common carotid artery were performed on both sides of the longitudinal projection immediately proximal to its bifurcation, using a 10 MHz linear transducer. The mean value of the right and left IMT was used for further analysis.

### 2.6. Transthoracic Echocardiogram

Additionally, a transthoracic echocardiogram (TTE) was performed for every participant with measurements of left ventricular (LV) ejection fraction (LVEF) and systolic pulmonary artery pressure (SPAP) using standard methods by two echocardiographers [26]. Other echocardiographic parameters, including left atrial volume, tricuspid regurgitation velocity, and mitral inflow pattern, were also recorded. Pulse-Doppler in a four-chamber apical view was used to measure peak E and A waves, flow velocity, and E/A ratio. Furthermore, the diastolic e’ velocity (at both the septal and lateral mitral origins) as well as filling indexes (E/e’ ratios) for the left ventricle (LV) were marked using tissue-Doppler imaging (TDI) in a four-chamber apical view. Finally, the echocardiographers assessed degenerative changes in aortic and/or mitral valves. They classified them into three groups: 1—no changes, 2—mild lesions, and 3—severe lesions, based on the severity of pathological changes.

### 2.7. Statistics

Statistical analyses were obtained using STATISTICA Tibco 13.3 software (StatSoft Inc., Tulsa, OK, USA). Categorical variables were presented as numbers (percentages), and differences between patient and control groups were compared using the Chi^2^ test. The Shapiro–Wilk test evaluated data distribution. The continuous variables were provided as medians with interquartile range or mean with 95% confidence interval (CI) and compared using the Mann-Whitney U test, Kruskal–Wallis, or unpaired t-tests appropriate. A one-way analysis of covariance (ANCOVA) was performed to adjust for potential confounders, including age, sex, BMI, smoking habit, family history of CVD, and comorbidities, such as hypertension, diabetes mellitus, and hypercholesterolemia. A Pearson correlation coefficient was used to analyze the associations between continuous variables. According to the severity of degenerative aortic/mitral valve changes, Kruskal–Wallis and multiple comparison tests with Hochberg correction were used in the subgroup analysis. Multiple linear regression models, built by a forward stepwise selection procedure (verified by the Snedecor’s F-distribution, with F > 1), were established to find independent determinants of FMD% and IMT. The R^2^ was checked to measure the variance. Receiver Operating Characteristic (ROC) curves were established to determine optimal cut-off values of FMD% and IMT and calculate odds ratio (OR) with a 95% CI. The statistical significance was set at a *p*-value lower than 0.05.

## 3. Results

### 3.1. Characteristics of Patients and Controls

The demographic, clinical, and laboratory characteristics, including cardiovascular risk factors and basic laboratory tests in CRAO patients and controls, have been presented in Table 1. Both groups were similar according to age, sex, and BMI. However, the CRAO group was characterized by a higher prevalence of comorbidities, including systemic hypertension, diabetes mellitus, and hypercholesterolemia. Furthermore, patients with CRAO were more frequently active smokers and had a positive family history of CVD.

Altogether, 88.1% (*n* = 111) of the CRAO group had at least one out of five previously mentioned cardiovascular risk factors. Additionally, systolic and diastolic blood pressure were significantly higher in the CRAO patients (for systolic BP, defined as values above the cut-off point of 140 mmHg, OR = 2.75 [95% CI, 2.01–3.76]; for diastolic BP, defined as values above the cut-off point of 90 mmHg; OR = 1.83 [95% CI, 1.35–2.50]). As expected, CRAO subjects also frequently received internal medicine medications, such as angiotensin-converting enzyme inhibitors or angiotensin receptor antagonists, used by 60 (48%) patients, diuretics by 35 (28%), beta-blockers by 29 (23%), and calcium channel blockers by 25 (20%) of them. Moreover, 25 (20%) patients were treated regularly with statins, and 45 (36%) received aspirin.

### 3.2. Ophthalmic Examination

In patients with CRAO, the duration of ocular symptoms (painless, sudden, and profound visual loss) ranged from 2 h to 5 days. At the presentation, the best-corrected visual acuity (BCVA) was between light perception and hand motions, and intraocular pressure ranged from 12 to 22 mmHg (mean: 17.4 mmHg). The relative afferent pupillary defect was profound in all patients. In all cases, the fundoscopy revealed a pale fundus with the “cherry-red spot” appearance of the macula and the attenuation of the retinal arterioles with the segmentation of blood flow. The optic disc was normal.

### 3.3. Basic Laboratory Tests

As shown in Table 1, CRAO patients had significantly higher CRP and homocysteine concentrations in their peripheral blood. They were also characterized by slightly increased hemoglobin levels and white blood cell counts. Furthermore, the CRAO group had elevated total cholesterol and triglycerides and lower high-density lipoprotein cholesterol levels. Additionally, we documented higher serum creatinine concentrations in patients.

### 3.4. Flow-Mediated Dilatation of the Brachial Artery (FMD) and Intima-Media Thickness of the Common Carotid Artery (IMT)

The detailed results of FMD% and IMT in the CRAO group and controls are provided in Figure 1 and Table 2.

The CRAO patients were characterized by a 38.2% decrease in FMD% compared with the control group (*p* < 0.001, also after adjustment for potential confounders) (Figure 1a, Table 2). In addition, they had a 2.90 (95% CI, 2.16–3.92) higher OR of having lower FMD%, defined as values below the cut-off point of 7.89% in comparison to controls.

Interestingly, in both groups, FMD% was inversely related to age (r = −0.33, *p* < 0.001 and r = −0.25, *p* = 0.01, for CRAO and control, respectively) and in controls to BMI (r = −0.25, *p* = 0.01). In patients, it was also lower in men (5.0% [95%CI, 4.08–6.67%], *n* = 63 vs. 6.8% [95%CI, 4.88–8.70%], *n* = 57, *p* = 0.001).

Among laboratory parameters, in CRAO we documented a negative association of FMD% with blood levels of creatinine and CRP (r = −0.18, *p* = 0.04 and r = −0.19, *p* = 0.04, respectively), whereas in controls, only with hemoglobin (r = −0.23, *p* = 0.02).

Surprisingly, internal comorbidities, smoking status, and a positive history of CVD in the family did not influence FMD% in the CRAO group. However, FMD% values in patients were lower in those who were receiving angiotensin-converting enzyme inhibitors/angiotensin receptor antagonists (5.29% [95% CI, 4.54–6.05%], *n* = 57 vs. 6.74% [95% CI, 6.03–7.44%], *n* = 63; *p* = 0.003) and calcium channel blockers (4.69% [95% CI, 3.54–5.84%], *n* = 24 vs. 6.39% [95% CI, 5.81–6.97%], *n* = 96; *p* = 0.006).

In a multiple regression model, unfavorable lower FMD% was independently determined by higher age and BMI, elevated CRP, increased tricuspid regurgitation velocity, suggesting higher SPAP, and a thicker interventricular septum (Table 3). On the contrary, higher LVEF positively predicted FMD% in this model. Nevertheless, all the mentioned variables explained only 33% of the FMD% variability.

Likewise, for FMD%, CRAO was related to an unfavorable 23.1% elevated IMT compared to the control group (*p* < 0.001, after adjustment for potential confounders) (Figure 1b, Table 2). The patient group was characterized by a 2.84 (95% CI, 2.07–3.89) increased OR of having elevated IMT, defined as values above the cut-off point of 0.78 mm.

Interestingly, in controls but not in patients, IMT was positively associated with age (r = 0.40, *p* < 0.001) and BMI (r = 0.26, *p* = 0.009). On the contrary, in the CRAO group, it remained in positive correlation with systolic and diastolic blood pressure (r = 0.19, *p* = 0.03; r = 0.27, *p* = 0.002, respectively), but also with glucose (r = 0.36, *p* < 0.001) and total cholesterol levels (r = 0.23, *p* = 0.01). As expected, according to CRAO patients, IMT was higher in smokers (0.97 mm [0.92–1.02 mm], *n =* 47 vs. 0.75 mm [0.72–0.79 mm], *n* = 79; *p* < 0.001), and in those with diabetes (1.05 mm [0.99–1.11 mm], *n* = 24 vs. 0.79 mm [0.76–0.83 mm], *n =* 88; *p* < 0.001).

A multiple regression model in the CRAO group demonstrated higher blood pressure, elevated total cholesterol and glucose in serum, and increased interventricular septum thickness, predicting unfavorable thicker IMT (Table 3). Surprisingly, higher inflammatory markers, such as white blood cell count and CRP, negatively influenced the IMT in this model. All of the listed above parameters explained 35% of the variability in IMT.

Finally, as expected, FMD% and IMT remained in an inverse relationship in both CRAO patients and controls (r = −0.22, *p* = 0.02 and r = −0.24, *p* = 0.01, respectively).

### 3.5. Basic Transthoracic Echocardiographic Parameters

The results of TTE parameters are provided in Table 4. The transthoracic echocardiographic measurements revealed that CRAO patients had a slightly lower LVEF and abnormal LV diastolic and systolic diameters than controls. They also had thickened LV walls and increased the mean diameter of the right ventricular with higher SPAP. Furthermore, the area and the left atrium volume were greater in the CRAO group. Considering tissue Doppler parameters, CRAO patients were characterized by elevated septal, lateral, and mean ratio of E/e’, and significantly lower septal and lateral e’ wave values. All in all, those indices suggested LV diastolic cardiac dysfunction. The LVEF of CRAO patients was negatively correlated to the diastolic and systolic diameters of the LV (r = −0.52, *p* < 0.001; r = −0.75, *p* < 0.001, respectively). Additionally, the thickness of the posterior LV walls was associated with left atrium indices, including diameter (r = 0.50, *p* < 0.001), area (r = 0.31, *p* < 0.001) and volume (r = 0.39, *p* < 0.001). In the patient group, CRP level was positively correlated with the thickness of the interventricular septum and the posterior LV wall (r = 0.21, *p* = 0.019 and r = 0.21, *p* = 0.021, respectively).

Interestingly, in patients with CRAO, degenerative and atherosclerotic changes of the aortic/mitral valves were found significantly more often than in controls. Proper structures were documented in 45 (35.7%)/85 (79.4%) subjects, mild lesions in 55 (43.7%)/19 (17.8%), and severe lesions in 26 (20.6%)/3 (2.8%) individuals, in the CRAO group/controls, respectively (*p* < 0.001, all). Of note, in the CRAO group, the severity of degenerative valve changes was associated with selected ultrasound parameters, including FMD%, IMT, interventricular septum and left ventricle posterior wall thickness, and a left atrium diameter. For more details, see Table 5.

### 3.6. Associations of the Flow-Mediated Dilatation of the Brachial Artery (FMD) and Intima-Media Thickness of the Common Carotid Artery (IMT) with Basic Transthoracic Echocardiographic Parameters

In the CRAO group, unfavorable decreased FMD% and thicker IMT remained in association with echocardiographic parameters describing increased heart chambers and their remodeled thicker walls. For example, FMD% values correlated inversely with the interventricular septum thickness (r = −0.41, *p* < 0.001), posterior wall thickness (r = −0.38, *p* < 0.001), right ventricular mid diameter (r = −0.24, *p* = 0.01), and left atrium diameter (r = −0.29, *p* = 0.001), area (r = −0.26, *p* = 0.005), and volume (r = −0.29, *p* = 0.002). Furthermore, there was a positive relationship between the IMT and interventricular septum thickness (r = 0.39, *p* < 0.001), posterior wall thickness (r = 0.34, *p* < 0.001), and the left atrium diameter (r = 0.30, *p* = 0.001) and volume (r = 0.22, *p* = 0.015).

## 4. Discussion

The present study showed that CRAO patients had endothelial dysfunction characterized by reduced FMD% of the brachial artery and thicker IMT of the carotid artery, indicating atherosclerotic arterial wall alteration. Furthermore, some echocardiographic parameters, such as chamber dimensions, LV wall thickness, and diastolic function, showed several abnormalities compared to controls. At the same time, however, internal medicine comorbidities, including arterial hypertension, diabetes mellitus, hypercholesterolemia, or traditional CVD risk factors, such as smoking, were documented much more frequently in the CRAO group. That observation is in accordance with the study performed by Callizo et al. [10], in which almost 80% of CRAO individuals had at least one new, previously undiagnosed CVD risk factor.

So far, FMD% in CRAO has not been evaluated. Nevertheless, some studies have demonstrated endothelial dysfunction expressed by a lower FMD in patients with retinal vascular obstruction other than CRAO [32,33]. In one of them, Gouliopoulos et al. [33] reported a reduction in FMD% in those with retinal vein occlusion. Since our study is the first in patients with CRAO, this underlines the novelty of the following research.

As expected, in our data, FMD% was determined by essential demographic factors, including sex, age, and BMI. This observation aligns with other reports, indicating FMD% predictors in several diseases [34,35]. Additionally, among laboratory parameters, the CRP level had an independent impact on FMD% in CRAO. This observation suggests a low-grade systemic inflammation involved in the CRAO pathogenesis since almost all CRP values in our patients were within the normal range. Inflammation favors the development of atherosclerosis and vascular dysfunction [36]. Premature atherosclerosis has been documented in patients with chronic inflammatory diseases, including asthma [34], antiphospholipid syndrome [37], or systemic lupus erythematosus [38]. Furthermore, a study by Goldenberg-Cohen et al. [39] showed that although CRP was not significantly elevated in CRAO patients, it correlated with the degree of atherosclerosis examined during the Doppler examination of the carotid arteries. However, these authors recruited only 16 patients with CRAO; thus, the results may be unreliable. In turn, a meta-analysis by Huang et al. [40] revealed that homocysteine seemed to be an independent risk factor of retinal artery occlusion (RAO), which stays in line with our study, at least to some extent.

In turn, lower FMD% in patients taking internal medicines such as angiotensin-converting enzyme inhibitors/angiotensin receptor antagonists and calcium channel blockers is an intriguing result and merits comment. We believe that those with prescribed medicaments suffered from a hard time controlling and treating hypertension; thus, they had more severe endothelium injury and, as a result, decreased FMD%.

In our data, CRAO patients had significantly elevated IMT of the carotid artery, indicating arterial wall atherosclerotic alteration. Other researchers have also demonstrated this observation [41,42]. Additionally, Marin-Sanabria et al. [41] documented more carotid plaques and carotid artery stenosis in patients with different types of vascular retinopathies, including eight subjects with retinal artery occlusion. Our study, however, is much larger, examining more than one hundred subjects, and it mutually analyzes FMD%, a recommended endothelial dysfunction biomarker.

Interestingly, in our research, IMT was related to age and BMI, but only in controls. On the contrary, in CRAO patients, unfavorable thicker IMT was determined by blood pressure, diabetes, smoking habit, serum glucose, and cholesterol levels, emphasizing the role of modifiable traditional CVD risk factors on atherosclerotic alterations. It has been shown that effective pharmacological treatment of hypertension and hypercholesterolemia may cause regression of atherosclerotic lesions, as depicted by a decrease in the IMT values [43]. Thus, lifestyle changes, e.g., smoking cessation, physical activity, diet modification, and proper therapy, including statins and more intensive hypotension and diabetes control, would also be helpful in the prevention of CRAO. Moreover, in the analysis of the lipid profile, CRAO patients had slightly higher levels of total cholesterol and triglycerides, and decreased HDL cholesterol levels. It is worth noting that a low level of HDL cholesterol is perceived as an independent risk factor for RAO development [44]. Interestingly, Chien et al. [45] have shown in the prospective study of a large group of patients that at least three months of statin therapy reduces the cumulative incidence rate of retinal vascular occlusion, including CRAO. Thus, monitoring lipid abnormalities seems essential in treating and preventing CRAO. Additionally, modern research on cilostazol revealed that this selective phosphodiesterase III inhibitor limits the progression of atherosclerosis also in CRAO patients [46]. However, more extensive observational and experimental studies are needed to recommend that medication for clinical use.

The final issue for discussion is the echocardiography results. CRAO patients had enlarged heart chambers, thicker LV walls, and impaired diastolic LV function. Interestingly, those abnormalities were associated with lower FMD% and thicker IMT, suggesting mutual remodeling changes in the heart structures and arterial walls. In addition, the patient group had increased SPAP, which may be related to many causes [47], e.g., LV hypertrophy, LV diastolic dysfunction, and smoking habits that promote airway inflammation and the development of chronic obstructive pulmonary disease. Unfortunately, spirometry was not a part of our study, and our patients were not screened for that disease. Diastolic cardiac dysfunction is diagnosed in many cardiological conditions, including arterial hypertension, aortic stenosis, and coronary artery disease [48]. However, it might also be related to coronary microcirculation dysfunction, referred to as lower FMD%. In addition, endothelial damage gives rise to cardiomyocyte mechanisms, leading to their hypertrophy and increased collagen production in the interstitial tissue, thus, wall remodeling [49,50]. Although LV diastolic dysfunction is usually clinically insignificant in asymptomatic subjects, it may increase the risk of cardiovascular events, including CRAO.

Finally, FMD% and IMT were inversely correlated in both analyzed groups in our study. Thus, both might be recommended as simple screening methods to evaluate endothelial damage in asymptomatic subjects threatened with a stroke or CRAO, particularly in those with coexisting essential CVD risk factors. A healthy lifestyle and intensive pharmacological prophylaxis of hypertension, lipid disorders, diabetes, and low doses of aspirin may be helpful in preventing such serious complications in these patients.

## 5. Limitations

Our study has several limitations. Firstly, CRAO patients had comorbidities, including hypertension, diabetes mellitus, and hypercholesterolemia, likely affecting endothelium and heart function. Nevertheless, the primary outcomes, such as FMD% and IMT, were adjusted for those confounders using adequate statistical methods (ANCOVA). Thus, we believe that the comorbidities did not affect the final results. Secondly, FMD% and IMT are, to some extent, subjective. Therefore, they were measured by two independent and experienced sonographers, blinded according to the case and control, and the final results were presented as an arithmetic mean to reduce the error. Furthermore, some of the presented associations may be incidental and not represent a cause-and-effect relationship. Eventually, largely observational and prospective studies are needed to verify our findings; however, it is worth emphasizing that in this study, the CRAO group of patients is the largest one that has been analyzed till now.

## 6. Conclusions

Our study showed that CRAO patients demonstrated vascular endothelial injury, thicker IMT, and diastolic LV cardiac dysfunction, likely related to the increased risk of cardiovascular events. Performed measurements are non-invasive and straightforward; thus, we recommend them as a valuable diagnostic tool in identifying asymptomatic subjects with an increased risk of CRAO development. As CRAO may herald the onset of severe and even life-threatening cardiovascular events, these patients require an immediate systemic approach and initiate the necessary systemic treatment to minimize the risk of future vascular complications.

## Figures and Tables

**Figure 1 jcm-11-02263-f001:**
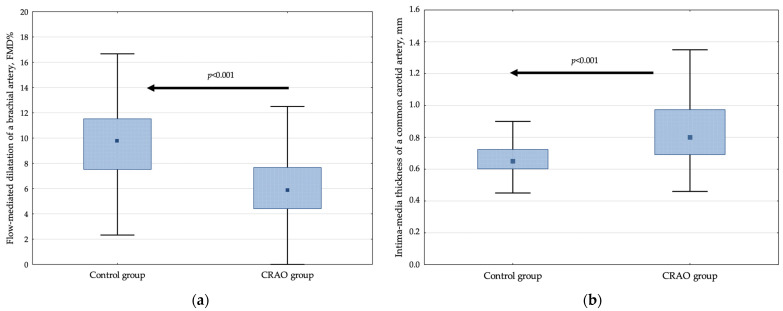
The relative increase in flow-mediated dilatation of the brachial artery (**a**) and mean values of intima-media thickness of the common carotid artery (**b**) in controls and CRAO patients. Data are presented as median, interquartile range, and maximum and minimum values. The numbers on the graph represent *p*-values compared with the control group.

**Table 1 jcm-11-02263-t001:** A summary of demographic, clinical, and laboratory characteristics in central retinal artery occlusion patients and a control group.

Parameter	Patients *n* = 126	Controls *n* = 107	*p*-Value
**Demographic parameters**			
Age, years	55.7 (53.7–57.7)	53.7 (51.9–55.5)	0.1
Sex, male, *n* (%)	66 (52.4%)	56 (52.3%)	0.9
Body mass index, kg/m^2^	26.6 (25.5–28.3)	26.4 (24.2–27.9)	0.1
**Clinical characteristics**			
Systolic blood pressure, mmHg	140 (130–145)	130 (120–135)	<0.001 *
Diastolic blood pressure, mmHg	85 (80–90)	80 (70–85)	<0.001 *
Hypertension, *n* (%)	49 (38.9%)	27 (25.2%)	0.038 *
Diabetes mellitus, *n* (%)	24 (19.0%)	9 (8.4%)	0.012 *
Hypercholesterolemia, *n* (%)	85 (67.5%)	46 (43.0%)	<0.001 *
Smoking habit, *n* (%)	47 (37.3%)	23 (21.5%)	0.02 *
Positive family history of CVD, *n* (%)	46 (36.5%)	20 (18.7%)	0.004 *
**Basic laboratory tests**			
Blood platelets, 10^3^/μL	228 (190–276)	228 (198–278)	0.86
Hemoglobin, g/dL	14.5 (13.6–15.4)	13.9 (12.8–14.8)	<0.001 *
White blood cells, 10^3^/μL	6.8 (5.3–7.9)	6.1 (5.0–7.1)	0.01 *
Total cholesterol, mmol/L	5.4 (4.9–5.9)	5.0 (4.6–5.5)	<0.001 *
HDL cholesterol, mmol/L	1.2 (1.1–1.4)	1.4 (1.2–1.6)	<0.001 *
LDL cholesterol, mmol/L	3.2 (2.9–3.7)	3.1 (2.8–3.4)	0.13
Triglycerides, mmol/L	1.8 (1.4–2.1)	1.7 (0.9–1.9)	<0.001 *
Glucose, mmol/L	5.7 (4.9–6.1)	5.4 (4.9–5.8)	0.10
Creatinine, μmol/L	88.3 (78.0–98.0)	79.2 (69.3–92.1)	<0.001 *
C-reactive protein, mg/L	4.8 (3.6–6.5)	2.8 (1.8–3.9)	<0.001 *
Homocysteine, μmol/L	13.1 (10.2–15.2)	10.6 (9.4–13.5)	<0.001 *

Categorical variables are presented as numbers (percentages), continuous variables as median and interquartile range or mean with 95% confidence interval, as appropriate. The statistically significant results are marked *. Abbreviations: *n*—number, CVD—cardiovascular disease, HDL—high-density lipoprotein, LDL—low-density lipoprotein.

**Table 2 jcm-11-02263-t002:** Flow-mediated dilatation of a brachial artery and intima-media thickness of a common carotid artery in patients with central retinal artery occlusion and controls.

Parameter	Patients *n* = 126	Controls *n* = 107	*p*-Value
**Ultrasound parameters of endothelial injury and atherosclerosis**			
Relative increase in flow-mediated dilatation of a brachial artery, %	5.88 (4.40–7.69)	9.52 (7.50–11.36)	<0.001 *
Mean value of intima-media thickness of a common carotid artery, mm	0.80 (0.69–0.98)	0.65 (0.58–0.73)	<0.001 *

Continuous variables as median and interquartile range or mean with 95% confidence interval, as appropriate. The statistically significant results are marked *. Abbreviations: *n*—number.

**Table 3 jcm-11-02263-t003:** Results of multiple linear regression models for a relative increase in flow-mediated dilatation (FMD%) of the brachial artery and intima-media thickness (IMT) of the common carotid artery in the central retinal artery occlusion patients.

	β (95% CI)	R^2^	Adjustment Statistics
**Relative increase in flow-mediated dilatation of the brachial artery, %**			
Age, years	−0.269 (−0.36 to −0.18)	0.33	F = 6.68, *p* < 0.00001
Body mass index, kg/m^2^	−0.137 (−0.23 to −0.05)		
C-reactive protein, mg/L	−0.155 (−0.25 to −0.06)		
Tricuspid regurgitation velocity, m/s	−0.253 (−0.35 to −0.16)		
Left ventricular ejection fraction, %	0.124 (0.03 to 0.22)		
Interventricular septum thickness, cm	−0.288 (−0.38 to −0.19)		
**Intima-media thickness of the common carotid artery, mm**			
Systolic blood pressure, mmHg	0.120 (0.04 to 0.20)	0.35	F = 7.95, *p* < 0.00001
White blood cells, 10^3^/μL	−0.112 (−0.20 to −0.02)		
Total cholesterol, mmol/L	0.174 (0.07 to 0.28)		
Glucose, mmol/L	0.260 (0.16 to 0.36)		
C-reactive protein, mg/L	−0.099 (−0.19 to −0.01)		
Interventricular septum thickness, cm	0.430 (0.34 to 0.52)		

The resulting standardized regression coefficient (β) with 95% confidence interval (95%CI) for a factor (independent variable) indicates the increase or decrease in standard deviations (SDs) of a dependent variable (FMD% or IMT), when that particular factor increases with 1 SD and all other variables in the model remain unchanged. Abbreviations: R^2^—a statistical measure of the quality of model fit.

**Table 4 jcm-11-02263-t004:** Basic transthoracic echocardiographic parameters in patients with central retinal artery occlusion and controls.

Parameter	Patients *n* = 126	Controls *n* = 107	*p*-Value
**Left ventricular basic parameters**			
LV ejection fraction, %	67 (65–69)	68 (66–70)	0.026 *
LV end-diastolic dimension, mm	49.5 (48.7–50.2)	47.8 (47.2–48.5)	0.008 *
LV end-systolic dimension, mm	31.0 (29.0–33.0)	30.0 (29.0–32.0)	0.011 *
LV posterior wall thickness, cm	1.1 (0.9–1.2)	1.0 (0.9–1.0)	<0.001 *
Interventricular septum thickness, cm	1.1 (1.0–1.4)	1.0 (0.9–1.1)	<0.001 *
**Left ventricular diastolic function**			
MV E-wave, cm/s	90 (80–100)	90 (80–95)	0.26
MV A-wave, cm/s	95 (70–110)	70 (70–88)	<0.001 *
MV E/A ratio, *n*	0.90 (0.82–1.22)	1.10 (0.80–1.28)	0.14
MV TDI septal e’, cm/s	9.0 (7.0–11.0)	11.0 (9.0–12.0)	<0.001 *
MV TDI lateral e’, cm/s	9.5 (7.5–12.0)	12.0 (10.0–13.0)	<0.001 *
MV TDI E/septal e’ ratio, *n*	10.0 (8.0–12.9)	8.1 (6.9–10.0)	<0.001 *
MV TDI E/lateral e’ ratio, *n*	9.3 (7.3–12.0)	7.4 (6.4–9.1)	<0.001 *
MV TDI mean value of E/e’ ratio, *n*	9.7 (7.6–12.4)	7.9 (6.7–9.6)	<0.001 *
**Right ventricular diameter**			
Right ventricular mid diameter, mm	22 (20–23)	21 (20–23)	0.004 *
**Left atrium parameters**			
LA diameter, cm	3.9 (3.7–4.2)	3.8 (3.6–4.0)	0.14
LA area, cm^2^	21.5 (19.5–23.5)	19.0 (17.8–21.0)	<0.001 *
LA volume, mL	34.2 (30.6–36.5)	30.3 (28.4–33.5)	<0.001 *
**Assessment of pulmonary hypertension**			
Pulmonary artery systolic pressure, mmHg	33.1 (31.9–34.3)	28.7 (27.7–29.6)	<0.001 *
Tricuspid regurgitation velocity, m/s	2.6 (2.4–2.9)	2.4 (2.2–2.7)	<0.001 *

Categorical variables are presented as numbers (percentages), continuous variables as median and interquartile range, or mean with 95% confidence interval, as appropriate. The statistically significant results are marked *. Abbreviations: *n*—number, LV—left ventricle, MV—mitral valve, TDI—tissue Doppler imaging, LA—left atrium.

**Table 5 jcm-11-02263-t005:** Characteristics of selected ultrasound parameters according to the severity of degenerative valve lesions.

Ultrasound Parameter	A Group with:			*p*-Value ^#^			
	No Changes (0)	Mild Changes (1)	Severe Changes (2)	All Groups	0–1	0–2	1–2
Relative increase in flow-mediated dilatation of a brachial artery, %	8.88 (8.33–9.43)	6.64 (6.00–7.27)	5.85 (3.65–8.05)	0.009 *	0.09	<0.001 *	0.018 *
Mean value of intima-media thickness of a carotid artery, mm	0.68 (0.66–0.70)	0.80 (0.76–0.85)	0.93 (0.85–1.01)	<0.001 *	0.028 *	<0.001 *	0.014 *
Interventricular septum thickness, cm	1.02 (1.00–1.05)	1.13 (1.09–1.17)	1.31 (1.23–1.38)	<0.001 *	0.015 *	<0.001 *	<0.001 *
LV posterior wall thickness, cm	0.98 (0.95–1.00)	1.06 (1.03–1.10)	1.18 (1.12–1.23)	<0.001 *	0.007 *	<0.001 *	0.006 *
LA diameter, cm	3.79 (3.73–3.86)	3.95 (3.87–4.03)	4.15 (4.00–4.29)	0.002 *	0.06	<0.001 *	0.004 *

Results are presented as mean with 95% confidence interval and these that reached statistical significance are marked *. ^#^—*p*-values after adjustment for potential confounders including age, sex, and body mass index, comparing all groups together; group (0) with (1); (0) with (2); (1) with (2), respectively. Abbreviations: LV—left ventricle, LA—left atrium.

## Data Availability

The data presented in this study are available on a reasonable request from the corresponding author.

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
