# Peer review of "Central Retinal Artery Occlusion Is Related to Vascular Endothelial Injury and Left Ventricular Diastolic Dysfunction"

_jcm, 2022, doi:10.3390/jcm11082263_

Round 1

Reviewer 1 Report

This study investigated the relationship between central retinal artery occlusion (CRAO) and cardiovascular function. A total of 126 consecutive CRAO patients and 107 control participants were included in the study. Several cardiovascular parameters were collected and the authors found increased dimensions of the left atrium and thicker left ventricular walls with impaired left ventricular diastolic function in CRAO patients. The authors conclude that non-invasive ultrasound assessments may be helpful in screening patients with increased CRAO risk.

    1. Please describe the diagnostic criteria of CRAO in detail. What’s the severity of CRAO in the patients recruited?
    2. Please describe the time interval between the onset of CRAO and cardiovascular studies performed.
    3. It’s well known that CRAO may be related to the systemic condition. Please describe the benefit of this study in instructing clinical practice. Is there any preventive medicine for these patients with high risk?

Reviewer 2 Report

Well designed study in which flow-mediated dilatation of the brachial artery, intima-media thickness of the common carotid artery and and transthoracic echocardiogram was performed in patients suffering from central retinal artery occlusion and normal subjects. The authors explain in a clear way some of the limitations such as comorbidities, hypertension, diabetes mellitus or hypercholesterolemia that could influence the results. I have no questions or comments to add.
